# Generation of Narrow Beams of Ultrarelativistic Positrons (Electrons) in the Breit–Wheeler Resonant Process Modified by the Field of a Strong Electromagnetic Wave

**Sergei P. Roshchupkin** *[ID], **Vitalii D. Serov** [ID] and **Victor V. Dubov**

Department of Theoretical Physics, Peter the Great St. Petersburg Polytechnic University, Polytechnicheskaya 29, 195251 Saint Petersburg, Russia; vitalii_serov@inbox.ru (V.D.S.); dubov@spbstu.ru (V.V.D.)
* Correspondence: serg9rsp@gmail.com

**Abstract:** The resonant external field-assisted Breit–Wheeler process (Oleinik resonances) for strong electromagnetic fields with intensities that are less than the critical Schwinger field that has been theoretically studied. The resonant kinematics were studied in detail. The case of high-energy initial gamma quanta and emerging ultrarelativistic electron–positron pairs was studied. The resonant differential cross section was obtained. The generation of narrow beams of ultrarelativistic positrons (for Channel A) and electrons (for Channel B) was predicted with a probability that significantly exceeded the corresponding nonresonant process.

**Keywords:** external field QED; Oleinik's resonances; Breit–Wheeler process

## 1. Introduction

Over the past several decades, there has been significant interest in studying the processes of quantum electrodynamics (QEDs) in external electromagnetic fields (see, for example, reviews [1–7], monographs [8–10], and articles [11–57]). This is mainly associated with the appearance of lasers with high radiation intensities and beams of small transverse dimensions [11–18].

An important feature of high-order regarding the fine structure constant of QED processes in an external field is the potential for their resonant occurrence, where virtual intermediate particles enter the mass shell. Such resonances were first considered by Oleinik [19,20]. Under resonance conditions, the conservation laws of energy and momentum are satisfied for intermediate particles in an external field. As a result, second-order processes by the fine structure constant effectively reduce into two sequential first-order processes. A detailed discussion of resonant processes is presented in reviews [2,4], monographs [8–10], as well as recent articles [28–31]. It is important to note that the probability of resonant processes can significantly exceed the corresponding probabilities of nonresonant processes.

The process of electron–positron pair production by two gamma quanta was first considered by Breit and Wheeler [32]. Currently, there is a significant number of works devoted to the study of the Breit–Wheeler process in an external electromagnetic field (see, for example, [33–44]). It should be noted that a distinction should be made between the external field-stimulated Breit–Wheeler process (a first-order process with respect to the fine structure constant) and the external field-assisted Breit–Wheeler process (a second-order process with respect to the fine structure constant). In this paper, Oleinik's resonances for the external strong field-assisted Breit–Wheeler process will be investigated. It should be noted that, in a weak field, this process was considered in the article [44]. It is important to note that, under the conditions of resonance and the absence of interference between different reaction channels, the original second-order process effectively reduces to two first-order processes: the external field-stimulated Breit–Wheeler process and the external field-stimulated Compton effect [44].

Breit–Wheeler pair production (a nonresonant process) yielded by two high-frequency photons in the presence of a low-frequency field has also been considered in articles [45,46]. The generation of highly collimated ultrarelativistic positron beams through laser-driven pair production has also been studied in articles [47–49]. Currently, there are concrete plans in various strong-field laboratories to measure the nonlinear Breit–Wheeler effect [50–52].

The main parameter for describing the Breit–Wheeler process in the field of a plane electromagnetic wave is the classical relativistic invariant parameter:

$$\eta = \frac{eF\lambda}{mc^2}, \tag{1}$$

which is numerically equal to the ratio of the work of the field on the wavelength to the rest energy of the electron. Here, $e$ and $m$ are the charge and mass of the electron, respectively, $F$ and $\lambda = c/\omega$ are the electric field strength and wavelength, respectively, and $\omega$ is the frequency of the wave [1].

In this paper, we consider the resonant strong electromagnetic field-assisted Breit–Wheeler process for high-energy gamma quanta with energies of $\hbar\omega_{1,2} \lesssim 10^2$ GeV. Therefore, we will consider high-energy gamma quanta in the following, thereby ensuring that the produced electron–positron pair in a field of the wave is ultrarelativistic.

$$\hbar\omega_{1,2} \gg mc^2, \quad E_\pm \gg mc^2. \tag{2}$$

Here, $\hbar\omega_{1,2}$ and $E_\pm$ are the energies of the initial gamma quanta and the final positron or electron, respectively. Therefore, we will assume that the magnitude of the classical parameter $\eta$ is upper bounded by the following condition:

$$\eta \ll \eta_{max}, \quad \eta_{max} = \min\left(\frac{E_\pm}{mc^2}\right). \tag{3}$$

Let us estimate the maximum intensity of the electric field in the wave. For electron–positron pair energies of $E_\pm \lesssim 10^2$ GeV, it follows from Equation (3) that $\eta \ll \eta_{max} \sim 10^5$, or, for the field strength, we have $F \ll F_{max} \sim 10^{15}$ Vcm$^{-1}$ ($I \ll I_{\max} \sim 10^{28}$ Wcm$^{-2}$). Thus, the problem will consider sufficiently large intensities of the electromagnetic wave. However, these fields must be smaller than the Schwinger critical field $F_* \approx 1.3 \times 10^{16}$ Vcm$^{-1}$ [5,55].

In the following, the relativistic system of units is used: $c = \hbar = 1$.

## 2. Amplitude of the Process

Let us consider this process in the field of a plane that is a circularly polarized wave propagating along the z axis:

$$A(\varphi) = \frac{F}{\omega}\left(e_x \cos\varphi + \delta e_y \sin\varphi\right), \quad \varphi = (kx) = \omega(t - z), \quad \delta = \pm 1. \tag{4}$$

Here, $e_x$ and $e_y$ are the four polarization vectors of the external field that have the following properties: $e_x = (0, \mathbf{e_x})$, $e_y = (0, \mathbf{e_y})$, $e_x e_y = 0$, and $(e_x)^2 = (e_y)^2 = -1$. The external field-assisted Breit–Wheeler process is characterized by two Feynman diagrams (Figure 1).

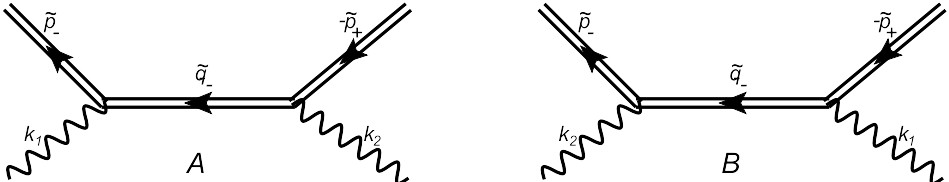

**Figure 1.** Feynman diagrams of electron–positron pair production by two gamma quanta in an external field, Channels A and B. External double lines correspond to Volkov functions of electron or positron, wavy lines correspond to wave functions of initial gamma quanta, and internal double lines correspond to Green's electron function in the field of a plane electromagnetic wave.

The amplitude of the considered process is written as follows:

$$S_{if} = ie^2 \iint d^4x_1 d^4x_2 \overline{\Psi}_{p_-}(x_1|A) \hat{A}_1(x_1; k_1) G(x_2 x_1|A) \hat{A}_2(x_2; k_2) \Psi_{-p_+}(x_2|A) + (k_1 \leftrightarrow k_2), \tag{5}$$

where $k_{1,2} = (\omega_{1,2}, \mathbf{k_{1,2}})$ respresents the four momenta of the initial gamma quanta, and $p_\pm = (E_\pm, \mathbf{p}_\pm)$ represents the four momenta of the final electron and positron. Here and further, the notation for the convolution of a 4-vector polarization with the Dirac gamma matrices is used: $\hat{A}_{1,2} \equiv \gamma_\mu A_{1,2}^\mu$ ; $\mu = 0, 1, 2, 3$. The four potentials of the initial gamma quanta $A_j$ in expression (5) are determined by the following functions:

$$A_j(x; k_j) = \sqrt{\frac{2\pi}{\omega_j}} \varepsilon_j e^{-ik_j x}, \quad j = 1, 2, \tag{6}$$

where $\varepsilon_j$ represents the four vectors of the polarization of the initial gamma quanta.

In the amplitude (5), the electron–positron pair corresponds to the Volkov functions [53,54]:

$$\Psi_p(x|A) = \mathfrak{J}_p(x) \frac{u_p}{\sqrt{2E}}, \quad \mathfrak{J}_p(x) = \left[ 1 + \frac{e}{2(pk)} \hat{k} \hat{A}(kx) \right] e^{iS_p(x)}, \tag{7}$$

and

$$S_p(x) = -(px) - \frac{e}{(kp)} \int_0^{kx} d\varphi [pA(\varphi) - \frac{e}{2} A^2(\varphi)], \tag{8}$$

where $u_p$ is the Dirac bispinor. The intermediate state in the amplitude (5) corresponds to the Green's function of the electron in the field of a plane wave $G(x_2 x_1|A)$ [56]:

$$G(x_2 x_1|A) = \int \frac{d^4 p}{(2\pi)^4} \mathfrak{J}_p(x_2) \frac{\hat{p} + m}{p^2 - m^2} \overline{\mathfrak{J}}_p(x_1). \tag{9}$$

After simple transformations, the amplitude (5) can be represented as follows:

$$S_{if} = \sum_{l=-\infty}^{+\infty} S_l, \tag{10}$$

where the partial amplitude $S_l$ corresponds to the absorption or emission of $|l|$ photons of the external wave. For the Channel A, the partial amplitude can be represented in the following form:

$$S_l = \frac{i\pi e^2 (2\pi)^4 e^{-id}}{\sqrt{\widetilde{E}_- \widetilde{E}_+ \omega_1 \omega_2}} [u_{p_-} M_l v_{p_+}] \delta^{(4)}(k_1 + k_2 - \widetilde{p}_- - \widetilde{p}_+ - lk). \tag{11}$$

Here, $d$ is the phase, which is independent of the summation indices, and $M_l$ represents the the matrix determined by the following expression:

$$M_l = \varepsilon_{1\mu}\varepsilon_{2\nu}\sum_{r=-\infty}^{+\infty} K_{l+r}^{\mu}(\widetilde{p}_-,\widetilde{q}_-)\frac{\hat{q}_- + m}{\widetilde{q}_-^2 - m_*^2}K_{-r}^{\nu}(\widetilde{q}_-,-\widetilde{p}_+), \quad \mu,\nu = 0,1,2,3. \tag{12}$$

In relation (12), the functions $K_{l+r}^{\mu}$ and $K_{-r}^{\nu}$ have the following form:

$$K_n^{\mu'}(\widetilde{p}',\widetilde{p}) = a^{\mu'}L_n(\widetilde{p}',\widetilde{p}) + b_-^{\mu'}L_{n-1} + b_+^{\mu'}L_{n+1}. \tag{13}$$

Here, the matrices $a^{\mu'}$ and $b_{\pm}^{\mu'}$ have the following form:

$$a^{\mu'} = \gamma^{\mu'} + \frac{m^2\hat{k}}{2(k\widetilde{p}')(k\widetilde{p})}k^{\nu}, \quad b_{\pm}^{\mu'} = \frac{1}{4}\eta m\left(\frac{\hat{e}_{\pm}\hat{k}\gamma^{\mu'}}{(k\widetilde{p}')} + \frac{\gamma^{\mu'}\hat{k}\hat{e}_{\pm}}{(k\widetilde{p})}\right), \tag{14}$$

and

$$e_{\pm} \equiv e_x \pm ie_y, \quad \mu' = \mu,\nu, \quad n = l+r,-r, \quad \widetilde{p} = -\widetilde{p}_+,\widetilde{q}_-, \quad \widetilde{p}' = \widetilde{q}_-,\widetilde{p}_-. \tag{15}$$

In relations (12) and (13), there are special functions $L_n$ [3], which, in the case of the circular polarization of the wave, can be represented using Bessel functions with integer indices:

$$L_n(\widetilde{p}',\widetilde{p}) = \exp(-in\chi_{\widetilde{p}'\widetilde{p}})J_n(\gamma_{\widetilde{p}'\widetilde{p}}), \tag{16}$$

where is then denoted as the following:

$$\gamma_{\widetilde{p}'\widetilde{p}} = m\eta\sqrt{-Q_{\widetilde{p}'\widetilde{p}}^2}, \quad \tan\chi_{\widetilde{p}'\widetilde{p}} = \delta\frac{(Q_{\widetilde{p}'\widetilde{p}}e_y)}{(Q_{\widetilde{p}'\widetilde{p}}e_x)}, \quad Q_{\widetilde{p}'\widetilde{p}} = \frac{\widetilde{p}'}{(p'k)} - \frac{\widetilde{p}}{(pk)}. \tag{17}$$

In the expressions (11) and (12), $\widetilde{p}_{\pm} = (\widetilde{E}_{\pm},\mathbf{\widetilde{p}}_{\pm})$ and $\widetilde{q}_-$ are the four quasimomenta of the electron (positron) and intermediate electron, and $m_*$ is the effective mass of the electron in the field of a circularly polarized wave (4) [1,2,8,29]:

$$\widetilde{q}_- = k_2 + rk - \widetilde{p}_{\pm}, \tag{18}$$

$$\widetilde{p}_{\pm} = p_{\pm} + \eta^2\frac{m^2}{2(kp_{\pm})}k, \quad \widetilde{q}_- = q_- + \eta^2\frac{m^2}{2(kq_-)}k, \tag{19}$$

and

$$\widetilde{p}_{\pm}^2 = m_*^2, \quad m_* = m\sqrt{1+\eta^2}. \tag{20}$$

## 3. The Resonant Kinematics

Under resonance conditions, both an electron and a positron can be intermediate particles. Therefore, instead of two Feynman diagrams in the nonresonant case (see Figure 1), under resonance conditions we will have four Feynman diagrams (see Figure 2): These are Channels A and B, as well as Channels A′ and B′, which are obtained from Channels A and B by rearranging the initial gamma quanta ($k_1 \leftrightarrow k_2$). Each channel in the resonance conditions effectively decays into two first-order processes via the fine structure constant: the external field-stimulated Breit–Wheeler process (EFSBWP) and the external field-stimulated Compton effect (EFSCE) with intermediate electrons and positrons entering the mass shell:

$$\widetilde{q}_-^2 = m_*^2, \quad \widetilde{q}_+^2 = m_*^2. \tag{21}$$

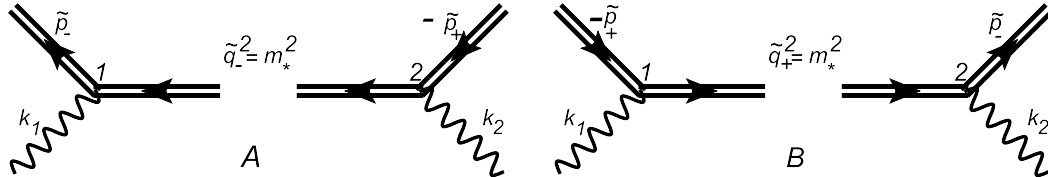

**Figure 2.** Feynman diagrams of the resonant electron–positron pair production by two gamma quanta in an external field for Channels A and B; to obtain Channels A' and B' replacement $k_1 \leftrightarrow k_2$ is needed.

Further considerations were carried out for the resonant Channels A and B (see Figure 2). It is important to emphasize that the laws of conservation of energy and of momentum for the intermediate processes of resonant Channels A and B have the following forms:

$$\text{EFSBWP}: \qquad k_2 + rk = \widetilde{q}_\mp + \widetilde{p}_\pm, \quad r = 1,2,3\ldots; \tag{22}$$

$$\text{EFSCE}: \qquad k_1 + \widetilde{q}_\mp = \widetilde{p}_\mp + r'k, \quad r' = 1,2,3\ldots \quad (r' = l + r). \tag{23}$$

Since the problem considers high-energy initial gamma quanta and ultrarelativistic energies of the final electron–positron pair (2), under such conditions, the momenta of the initial and final particles should lie within a narrow angle cone, which should be far away from the direction of the wave propagation [29,31] (see Figure 3):

$$\theta_{j\pm} \equiv \angle(\mathbf{k}_j, \mathbf{p}_\pm) \ll 1, \quad \theta_i \equiv \angle(\mathbf{k}_1, \mathbf{k}_2) \ll 1; \tag{24}$$

$$\theta \equiv \angle(\mathbf{p}_\pm, \mathbf{k}) \sim 1, \quad \theta_j \equiv \angle(\mathbf{k}_j, \mathbf{k}) \sim 1, \quad j = 1,2; \quad \theta \approx \theta_1 \approx \theta_2. \tag{25}$$

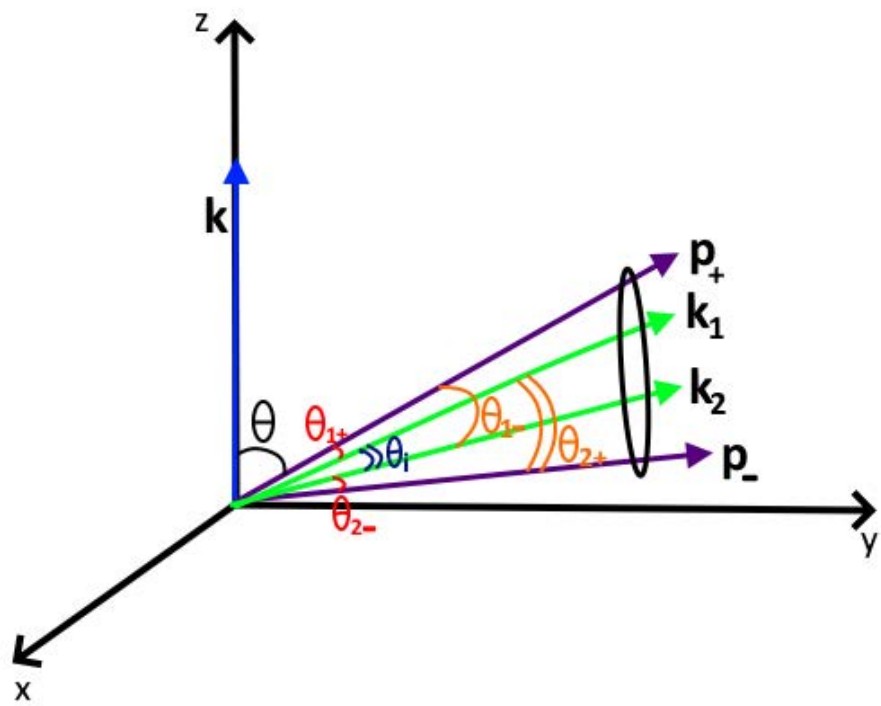

**Figure 3.** Geometry of the initial and final particles of the resonant Breit–Wheeler process.

Let us note that, under conditions (2) and (3), the expression for the positron (electron) quasienergy can be simplified:

$$\widetilde{E}_\pm = E_\pm \left[ 1 + \frac{1}{4\sin^2\frac{\theta}{2}} \left( \frac{m\eta}{E_\pm} \right)^2 \right] \approx E_\pm. \tag{26}$$

Let us determine the resonance energy of the positron (electron) for the second vertex (see Figure 2). By taking into account relations (2), (3), (21) and (24) from the conservation of the four momentum law (22) for the external field-stimulated Breit–Wheeler process, we obtain the resonance energies of the positron (for Channel A) or electron (for Channel B) in units of the total energy of the initial gamma quanta:

$$x_{j'(r)} = \frac{\omega_2}{2\omega_i(\varepsilon_{2BW(r)} + \delta_{2j'}^2)}\left[\varepsilon_{2BW(r)} \pm \sqrt{\varepsilon_{2BW(r)}(\varepsilon_{2BW(r)} - 1) - \delta_{2j'}^2}\right], \quad j' = +, -. \quad (27)$$

Here, the following is indicated:

$$x_{\pm(r)} = \frac{E_{\pm}(r)}{\omega_i}, \quad \omega_i = \omega_1 + \omega_2, \quad \delta_{2\pm} = \frac{\omega_2}{2m_*}\theta_{2\pm}. \quad (28)$$

In this case, the ultrarelativistic parameter $\delta_{2\pm}$, which determines the outgoing angle of the positron or electron, is contained within the following interval:

$$0 \le \delta_{2\pm}^2 \le \delta_{2max}^2, \quad \delta_{2max}^2 = \varepsilon_{2BW(r)}(\varepsilon_{2BW(r)} - 1). \quad (29)$$

It is important to emphasize that, in Equation (27), the quantity $\varepsilon_{2BW(r)}$ is bounded from below by the following unity:

$$\varepsilon_{2BW(r)} = r\varepsilon_{2BW} \ge 1, \quad \varepsilon_{2BW} = \frac{\omega_2}{\omega_{BW}}, \quad (30)$$

where $\omega_{BW}$ is the characteristic quantum energy of the following external field-stimulated Breit–Wheeler process [29,30]:

$$\omega_{BW} = \frac{m_*^2}{\omega \sin^2 \frac{\theta}{2}} = \begin{cases} 174\,\text{GeV} & \text{if} \quad \omega = 3\,\text{eV}, I = 1.675 \times 10^{19}\,\text{Wcm}^{-2} \\ 5.22\,\text{GeV} & \text{if} \quad \omega = 0.1\,\text{keV}, I = 1.861 \times 10^{22}\,\text{Wcm}^{-2} \\ 52.2\,\text{MeV} & \text{if} \quad \omega = 10\,\text{keV}, I = 1.861 \times 10^{26}\,\text{Wcm}^{-2}. \end{cases} \quad (31)$$

When estimating the value of the characteristic energy, the frequencies of the electromagnetic waves in the optical and X-ray ranges were used in Equation (31), as well as values of parameters $\eta = 1$ and $\theta = \pi$. It is worth noting that the ratio between the initial energy of the gamma quantum and the characteristic energy $\omega_{BW}$ determines the value of the parameter $\varepsilon_{2BW}$ (30), which can be either greater or less than unity. This significantly affects the number of photons absorbed in the EFBWP. Specifically, if the initial energy of the gamma quantum is less than the characteristic energy, then, as derived from Equations (30) and (31), it follows that this process occurs if the number of absorbed wave photons is above a certain minimum $r_{min}$ value, which is greater than unity:

$$r \ge r_{min} = \lceil \varepsilon_{2BW}^{-1} \rceil \quad (\omega_2 < \omega_{BW}). \quad (32)$$

If the initial energy of the gamma quantum is greater than the characteristic energy, then this process takes place already when one photon of the wave is absorbed:

$$r \ge 1 \quad (\omega_2 \ge \omega_{BW}). \quad (33)$$

Thus, the resonant energy of a positron (for Channel A) or an electron (for Channel B) is determined by two parameters: the corresponding outgoing angle of the positron ($\delta_{2+}^2$) or electron ($\delta_{2-}^2$), and the parameter $\varepsilon_{2BW(r)}$. At the same time, with a fixed parameter $\varepsilon_{2BW(r)}$, for each outgoing angle of the positron or electron, there are two possible energies (see Equation (27)).

Figure 4 shows the dependence of the energy of the positron (for Channel A) or electron (for Channel B) (see Equations (27)–(30)) for the external field-stimulated Breit–Wheeler process with the absorption of one and two photons of the wave at different frequencies, intensities of the electromagnetic wave (Equation (31), and various initial gamma quanta

energies. From this figure, it follows that the interval for the outgoing angle of the positron (electron) significantly depended on the number of absorbed photons of the wave. Additionally, for the same outgoing angle, there were two possible particle energies, except for the maximum outgoing angle.

Now, let us determine the resonant electron (positron) energy at the first vertex (see Figure 2). By taking into account Equations (2), (3), (21) and (24), from the conservation law of the four momentum (Equation (23)) of the external field-stimulated Compton effect, we obtained the resonant energies of the electron (for Channel A) or the positron (for Channel B) in terms of the total energy of the initial gamma quanta:

$$x_{\mp(r')} = \frac{\omega_1}{2\omega_i(\varepsilon_{1C(r')} - \delta_{1\mp}^2)}\left[\varepsilon_{1C(r')} + \sqrt{\varepsilon_{1C(r')}^2 + 4(\varepsilon_{1C(r')} - \delta_{1\mp}^2)}\right]. \tag{34}$$

Here, it is denoted as follows:

$$x_{\mp(r')} = \frac{E_{\mp}(r')}{\omega_i}, \quad \delta_{1\mp} = \frac{\omega_1}{m_*}\theta_{1\mp}. \tag{35}$$

$$\varepsilon_{1C(r')} = r'\varepsilon_{1C}, \quad \varepsilon_{1C} = \frac{\omega_1}{\omega_C}, \quad \omega_C = \frac{1}{4}\omega_{BW}. \tag{36}$$

Here, $\omega_C$ is the characteristic quantum energy of the external field-stimulated Compton effect [31]. This energy was four times less than the characteristic energy for the external field-stimulated Breit–Wheeler process. Additionally, it should be noted that the ultrarelativistic parameter $\delta_{1\mp}^2$, which determines the outgoing angle of the electron or positron, should not take values close to $\varepsilon_{1C(r')}$ in order to satisfy the condition $x_{\mp(r')} < 1$ (see Equation (34)). It should also be noted that there were no limitations on the parameter $\varepsilon_{1C(r')}$ for the external field-stimulated Compton effect. Therefore, this process occurred for any number of emitted photons of the wave $r' \geq 1$, whic is in contrast to the external field-stimulated Breit–Wheeler process, which had a threshold value for parameter $\varepsilon_{2BW(r)}$ (see Equations (30) and (32)).

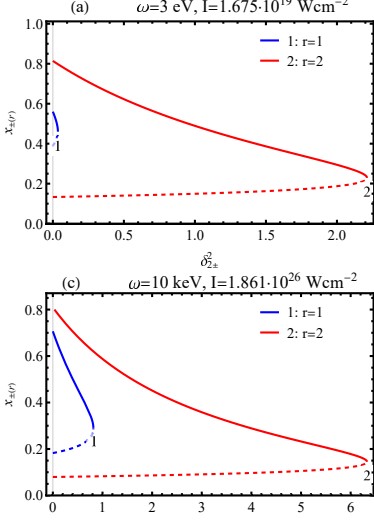
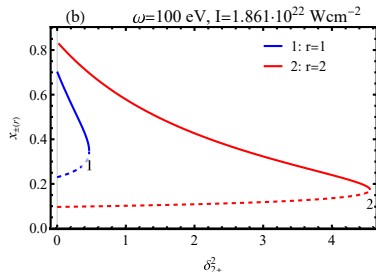

**Figure 4.** The energy of the positron (Channel A) or electron (Channel B) (27)–(30) for the external field-stimulated Breit–Wheeler process with the absorption of one and two photons of the wave at different frequencies and intensities of the electromagnetic wave (31). Solid lines correspond to the "+" signs, and dashed lines correspond to the "−" signs before the square root in (27). The energies of the initial gamma quanta were the following: (**a**) $\omega_1 = 10\,\text{GeV}, \omega_2 = 180\,\text{GeV}$; (**b**) $\omega_1 = 0.5\,\text{GeV}, \omega_2 = 7\,\text{Gev}$; and (**c**) $\omega_1 = 10\,\text{MeV}, \omega_2 = 80\,\text{MeV}$.

Furthermore, we assumed that the energies of the initial gamma quanta, within the framework of conditions (2), satisfied the additional conditions:

$$\omega_2 > \omega_{BW}, \quad \omega_1 \ll \omega_{BW}. \tag{37}$$

Conditions (37) represent the parameter $\varepsilon_{2BW} > 1$ and the parameter $\varepsilon_{1BW} \ll 1$ (see Equations (30) and (31)). Therefore, in Channels A and B, the external field-stimulated Breit–Wheeler process occurs with a number of absorbed photons of the wave $r \geq 1$, and, for the exchange resonant diagrams A′ and B′, the number of absorbed photons is $r \geq r_{min} = \lceil \varepsilon_{1BW}^{-1} \rceil \gg 1$. Thus, within the framework of conditions (37), the resonant Channels A′ and B′ would be suppressed, so we only considered two resonant Channels A and B (see Figure 2). It is also important to consider that, for Channel A, the resonant energy of the positron was determined by its outgoing angle relative to the momentum of the second gamma quantum in the EFBWP, while the resonant energy of the electron was determined by its outgoing angle relative to the momentum of the first gamma quantum in the EFSCE. For Channel B, we had the opposite situation, where the energy of the electron was determined by its outgoing angle relative to the momentum of the second gamma quantum, and the energy of the positron was determined by its outgoing angle relative to the momentum of the first gamma quantum (see Figure 2). Therefore, Channels A and B were distinguishable and did not interfere with each other.

It is important to note that, under resonance conditions (21), the resonant energies of the positron and electron for each reaction channel are determined by different physical processes: the external field-stimulated Breit-Wheeler process (27) and the Compton external field-stimulated effect (34). At the same time, the energies of the electron–positron pair are related to each other by the general law of the conservation of energy.

$$x_+ + x_- \approx 1 \quad (x_\pm = \frac{E_\pm}{\omega_i}). \tag{38}$$

It should be noted that, in Equation (38), we neglected a small correction term $|l|\omega/\omega_i \ll 1$. Taking into account Equations (27) and (34), as well as the law of conservation of energy (38) for Channels A and B, we obtained the following equations relating the outgoing angles of the positron and electron:

$$\delta_{1\mp}^2 = \varepsilon_{1C(r')} - \frac{(\omega_1/\omega_i)}{(1 - x_{\pm(r)})}\left[\varepsilon_{1C(r')} + \frac{(\omega_1/\omega_i)}{(1 - x_{\pm(r)})}\right]. \tag{39}$$

Here, the upper (lower) sign corresponds to Channel A (B). In Equation (39), the left side represents the ultrarelativistic parameter associated with the outgoing angle of the electron (positron) relative to the momentum of the first gamma quantum, and the right side is the function of the ultrarelativistic parameter $\delta_{2\pm}$ associated with the outgoing angle of the positron (electron) relative to the momentum of the second gamma quantum. Under the given parameters $\varepsilon_{1C(r')}$ and $\varepsilon_{2BW(r)}$, Equation (39) uniquely determines the outgoing angles of the electron and positron, and, therefore, their resonant energies (see Figures 4 and 5).

Figure 5 presents the dependence of the energy of the electron (for Channel A) or positron (for Channel B) (34), (39) for the external field-stimulated Compton effect at different frequencies, intensities of the electromagnetic wave (31), and initial gamma quanta energies under the condition of energy conservation in the first and second vertices (38). The graphs are given for different numbers of absorbed ($r$) and emitted ($r'$) photons of the wave.

It is also worth noting the important case when the quantum parameter was $\varepsilon_{2BW(r)} \gg 1$. In this case, as derived from the expression (27) with the "+" sign before the square root,

the energy of the positron (Channel A) or electron (Channel B) approached the energy of the highly energetic second gamma quantum:

$$E_{\pm} \approx \omega_2 \left[ 1 - \frac{(1 + 4\delta_{2\pm}^2)}{4\varepsilon_{2BW(r)}} \right] \longrightarrow \omega_2 \quad (\delta_{2\pm}^2 \ll \varepsilon_{2BW(r)}). \tag{40}$$

The expression with the "−" sign before the square root in Equation (27) leads to the minimum energy of the positron or electron $E_{\pm} \sim \omega_2 / \varepsilon_{2BW(r)} \ll \omega_2$. However, this case is unlikely. Similarly, for the first gamma quantum, when the quantum parameter was determined as $\varepsilon_{1C(r')} \gg 1$, we obtained that the energy of the electron (Channel A) or positron (Channel B) approached the energy of the first gamma quantum:

$$E_{\mp} \approx \omega_1 \left[ 1 - \frac{(1 + \delta_{1\mp}^2)}{\varepsilon_{1C(r')}} \right] \longrightarrow \omega_1 \quad (\delta_{1\mp}^2 \ll \varepsilon_{1C(r')}). \tag{41}$$

Thus, if the quantum parameters $\varepsilon_{1C(r')}$ and $\varepsilon_{2BW(r)}$ were to take large values, the resonant energies of the positron and electron would tend towards the energies of the corresponding initial gamma quanta.

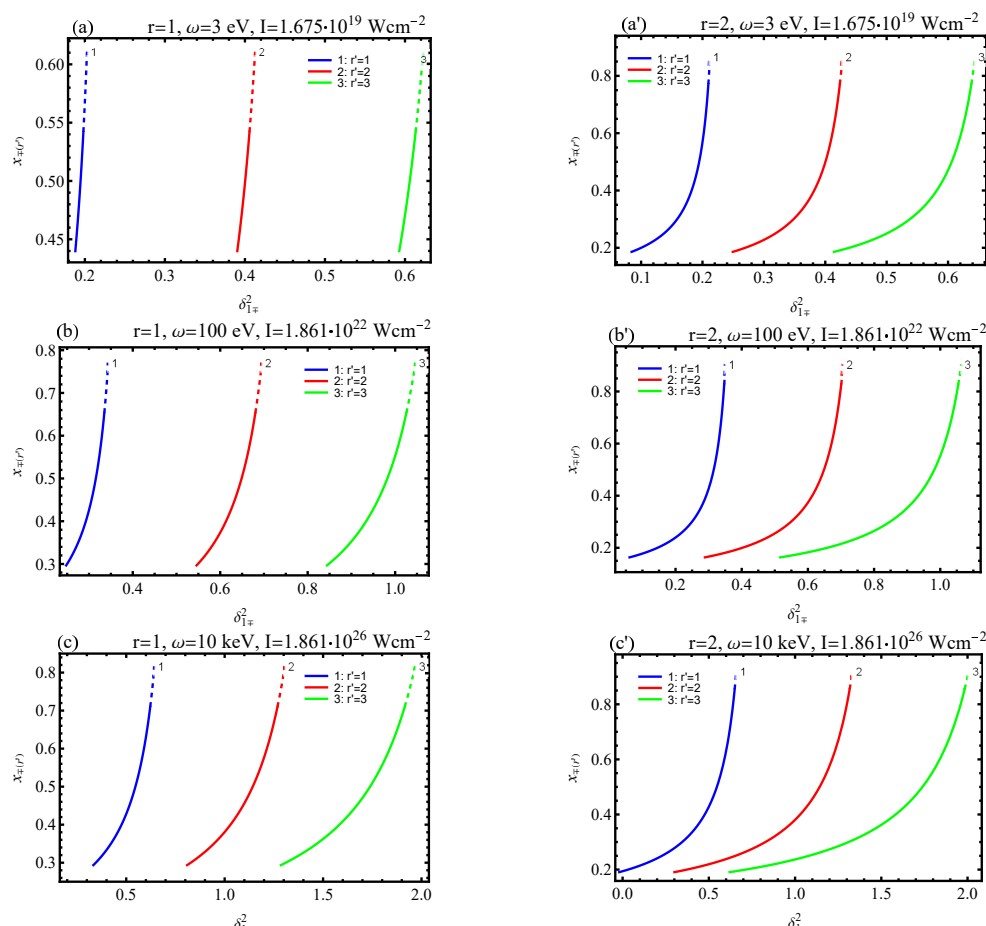

**Figure 5.** The dependence of the energy of the electron (Channel A) or positron (Channel B) (34), (39) for the external field-stimulated Compton effect at different frequencies, intensities of the electromagnetic wave (31), and initial gamma quanta energies under the condition of energy conservation in the first and second vertices (38). Solid lines correspond to the "+", and dashed lines correspond to the "−" signs before the square root in expressions (27), (34), and (39). The energies of the initial gamma quanta were the following: (**a**,**a′**) $\omega_1 = 10 \, \text{GeV}$, $\omega_2 = 180 \, \text{GeV}$; (**b**,**b′**) $\omega_1 = 0.5 \, \text{GeV}$, $\omega_2 = 7 \, \text{Gev}$; and (**c**,**c′**) $\omega_1 = 10 \, \text{MeV}$, $\omega_2 = 80 \, \text{MeV}$.

## 4. The Resonant Differential cross Section

Previously, it has been shown that, in conditions (2), (3), and (37), the exchange Channels A′ and B′ were suppressed. In addition, Channels A and B were distinguishable and, therefore, did not interfere (see text after Equation (37)). It is also important to note that the resonance processes with different numbers of absorbed and emitted wave photons corresponded to significantly different probabilities and energies of the final electron–positron pair. Therefore, they did not interfere either. Due to this, a summation of all the possible processes with absorption of *r* wave photons was not necessary in the amplitude (12).

$$M_{rr'} = \varepsilon_{1\mu}\varepsilon_{2\nu}K^{\mu}_{r'}(\widetilde{p}_-,\widetilde{q}_-)\frac{\hat{\widetilde{q}}_- + m}{\widetilde{q}^2_- - m^2_*}K^{\nu}_{-r}(\widetilde{q}_-, -\widetilde{p}_+), \quad r' = l + r. \tag{42}$$

The resonant differential cross section for Channels A and B, as well as the unpolarized initial gamma quanta and the final electron–positron pair, were obtained from the amplitude calculated from (10), (11) and (42) in a standard way [58]. After simple calculations, we obtained the following:

$$d\sigma_{rr'} = \frac{2m^6 r_e^2}{\widetilde{E}_-\widetilde{E}_+ m^2_*\delta^2_{\eta i}}\frac{K_{1\mp(r')}P_{2\pm(r)}}{|\widetilde{q}^2_\mp - m^2_*|^2}\delta^{(4)}\left[k_1 + k_2 - \widetilde{p}_- - \widetilde{p}_+ - (r'-r)k\right]d^3\widetilde{p}_-d^3\widetilde{p}_+. \tag{43}$$

Here, the upper (lower) sign corresponds to Channel A (B), and $r_e = e^2/m$ is the classical electron radius. In obtaining the resonant differential cross section (43), the resonant probability was divided by the flux density of the initial gamma quanta [58], whose equation is shown as follows:

$$j = \frac{(k_1 k_2)}{\omega_1\omega_2} \approx \frac{m^2_*}{2\omega_1\omega_2}\delta^2_{\eta i}, \quad \delta^2_{\eta i} \equiv \frac{\omega_1\omega_2}{m^2_*}\theta^2_i. \tag{44}$$

In expression (43), the function $P_{2\pm(r)}$ determines the probability of the external field-stimulated Breit–Wheeler process [1], and the function $K_{1\mp(r')}$ determines the probability of the external field-stimulated Compton effect [1], which is defined as follows:

$$P_{2\pm(r)} = J_r^2(\gamma_{2\pm(r)}) + \eta^2(2u_{2\pm(r)} - 1)\left[\left(\frac{r^2}{\gamma^2_{2\pm(r)}} - 1\right)J_r^2 + J_r'^2\right], \tag{45}$$

and

$$K_{1\mp(r')} = -4J_{r'}^2(\gamma_{1\mp(r')}) + \eta^2\left[2 + \frac{u^2_{1\mp(r')}}{1 + u_{1\mp(r')}}\right](J_{r'-1}^2 + J_{r'+1}^2 - 2J_{r'}^2). \tag{46}$$

The arguments of the Bessel functions for the external field-stimulated Breit–Wheeler process (45) and the external field-stimulated Compton effect (46) have the following forms:

$$\gamma_{2\pm(r)} = 2r\frac{\eta}{\sqrt{1+\eta^2}}\sqrt{\frac{u_{2\pm(r)}}{v_{2\pm(r)}}\left(1 - \frac{u_{2\pm(r)}}{v_{2\pm(r)}}\right)}, \tag{47}$$

and

$$\gamma_{1\mp(r')} = 2r'\frac{\eta}{\sqrt{1+\eta^2}}\sqrt{\frac{u_{1\mp(r')}}{v_{1\mp(r')}}\left(1 - \frac{u_{1\mp(r')}}{v_{1\mp(r')}}\right)}. \tag{48}$$

Here, the relativistic invariant parameters are equal to the following:

$$u_{1\mp(r')} = \frac{(k_1 k)}{(p_\mp k)} \approx \frac{(\omega_1/\omega_i)}{x_{\mp(r')}}, \quad v_{1\mp(r')} = \frac{2r'(q_\mp k)}{m^2_*} \approx \varepsilon_{1C(r')}\left(\frac{x_{\mp(r')}}{(\omega_1/\omega_i)} - 1\right), \tag{49}$$

and

$$u_{2\pm(r)} = \frac{(k_2 k)^2}{4(p_\pm k)(q_\mp k)} \approx \frac{(\omega_2/\omega_i)}{4x_{\pm(r)}\left(1 - \frac{x_{\pm(r)}}{(\omega_2/\omega_i)}\right)}, \quad v_{2\pm(r)} = r\frac{(k_2 k)}{2m_*^2} \approx \varepsilon_{2BW(r)}. \tag{50}$$

The elimination of the resonant singularity in expression (43) was carried out by the Breit–Wigner procedure [44,57]:

$$m_* \longrightarrow \mu_* = m_* - i\Gamma_{\mp(r)}, \quad \Gamma_{\mp(r)} = \frac{\widetilde{q}_\mp^0}{2m_*}W_1, \tag{51}$$

where $W_1$ is the total probability (per unit of time) of the external field-stimulated Compton effect on the intermediate electron (for Channel A) or positron (for Channel B) [1,31], which is derived through the following equations:

$$W_1 = \frac{\alpha m^2}{4\pi \widetilde{q}_\mp^0}K(\varepsilon_{1C}), \tag{52}$$

and

$$K(\varepsilon_{1C}) = \sum_{s=1}^{\infty} \int_0^{s\varepsilon_{1C}} \frac{du}{(1+u)^2}K(u, s\varepsilon_{1C}). \tag{53}$$

Here, $\alpha$ is the fine-structure constant, and the function $K(u, s\varepsilon_{1C})$ is determined by the following expression:

$$K(u, s\varepsilon_{1C}) = -4J_s^2(\gamma_{1(s)}) + \eta^2\left[2 + \frac{u^2}{1+u}\right](J_{s-1}^2 + J_{s+1}^2 - 2J_s^2) \tag{54}$$

$$\gamma_{1(s)} = 2s\frac{\eta}{\sqrt{1+\eta^2}}\sqrt{\frac{u}{s\varepsilon_{1C}}\left(1 - \frac{u}{s\varepsilon_{1C}}\right)}. \tag{55}$$

By taking into account the relations (51)–(55), the resonant denominator in the cross section (43) takes the following form:

$$|\widetilde{q}_\mp^2 - m_*^2|^2 \longrightarrow m_*^4\frac{x_{\mp(r')}^2}{(\omega_1/\omega_i)^2}\left[\left(\delta_{1\mp(0)}^2 - \delta_{1\mp}^2\right)^2 + Y_{\mp(r')}^2\right]. \tag{56}$$

Here, the ultrarelativistic parameter $\delta_{1\mp}^2$ is related to the resonance energy of the electron (for Channel A) or positron (for Channel B) by the relation (34), and the corresponding parameter $\delta_{1\mp(0)}^2$ can take arbitrary values that are unrelated to the energy of the electron (positron). In this case, the corresponding angular width of the resonance $Y_{\mp(r')}$ was determined by the following expression:

$$Y_{\mp(r')} = \frac{\alpha m^2}{4\pi m_*^2}\frac{\omega_1}{\omega_i x_{\mp(r')}}K(\varepsilon_{1C}). \tag{57}$$

Considering relation (26), we set $d^3\widetilde{p}_\pm \approx d^3 p_\pm$ and integrated the three-dimensional momentum of the electron (positron), as well as the energy of the positron (electron) for Channel A (for Channel B), using the delta-function in expression (43). After simple calculations, we obtained the following expression for the resonant differential cross section for Channels A and B:

$$R_{2\pm(rr')} = \frac{d\sigma_{rr'}}{d\delta_{2\pm}^2} = 8\pi r_e^2\left(\frac{m}{\delta_{\eta i}\omega_i}\right)^2\frac{x_{\pm(r)}}{x_{\mp(r')}^3}\left(\frac{m}{m_*}\right)^4\left(\frac{\omega_1}{\omega_2}\right)^2\frac{K_{1\mp(r')}P_{2\pm(r)}}{\left[\left(\delta_{1\mp(0)}^2 - \delta_{1\mp}^2\right)^2 + Y_{\mp(r')}^2\right]}. \tag{58}$$

Here, the upper (lower) sign corresponds to Channel A (B). It should be noted that the differential cross section (58) had a characteristic Breit–Wigner resonance structure [57]. Let us now determine the maximum resonant differential cross section:

$$\left(\delta_{1\mp(0)}^2 - \delta_{1\mp}^2\right)^2 \ll Y_{\mp(r')}^2. \tag{59}$$

Under conditions (59), the resonant cross section (58) takes its maximum value, which is equal to the following:

$$R_{2\pm(rr')}^{max} = \frac{d\sigma_{rr'}^{max}}{d\delta_{2\pm}^2} = r_e^2 c_{\eta i}\Psi_{\pm(rr')}. \tag{60}$$

Here, the function $c_{\eta i}$ is determined by the initial setup parameters

$$c_{\eta i} = \frac{2(4\pi)^3}{\alpha^2 K^2(\varepsilon_{1C})}\left(\frac{m}{\delta_{\eta i}\omega_2}\right)^2 \sim 10^8\left(\frac{m}{\delta_{\eta i}\omega_2}\right)^2, \tag{61}$$

and the functions $\Psi_{\pm(rr')}$ determine the spectral–angular distribution of the generated electron–positron pair:

$$\Psi_{\pm(rr')} = \frac{x_{\pm(r)}}{1 - x_{\pm(r)}}K_{1\mp(r')}P_{2\pm(r)}. \tag{62}$$

It is important to emphasize that the magnitude of the maximum resonant differential cross section significantly depends on the value of the function $c_{\eta i}$ (61). Let us require that the function $c_{\eta i} > 1$. Then, as derived from relation (61), we obtain a condition for the initial ultrarelativistic parameter $\delta_{\eta i}^2$ (44):

$$\delta_{\eta i}^2 < \left(10^4\frac{m}{\omega_2}\right)^2. \tag{63}$$

It should be noted that the corresponding Breit–Wheeler differential cross section without an external field in the kinematics (24) has the following order of magnitude [32]:

$$R_{BW} = \frac{d\sigma_{BW}}{d\delta_{2\pm}^2} \sim r_e^2\left(\frac{m}{\delta_i\omega_i}\right)^2, \quad \delta_i = \frac{\sqrt{\omega_1\omega_2}\theta_i}{m}. \tag{64}$$

As derived from relations (60)–(62) and (64), it can be seen that the maximum resonant cross section significantly exceeded the corresponding Breit–Wheeler cross section without an external field. Indeed, we estimated the value of the Breit–Wheeler differential cross section without an external field $R_{BW}$ (64). Thus, for $\delta_{\eta i} = 10^{-2}$ and different energies of the initial gamma quanta (see Table 1) we derived the following: if $\omega_1 = 10$ MeV, $\omega_2 = 80$ MeV, then $R_{BW} \sim 10^{-1}\,r_e^2$; if $\omega_1 = 0.5$ GeV, $\omega_2 = 7$ GeV, then $R_{BW} \sim 10^{-5}\,r_e^2$; and if $\omega_1 = 10$ GeV, $\omega_2 = 180$ GeV, then $R_{BW} \sim 10^{-7}\,r_e^2$.

Figure 6 shows the dependencies of the maximum resonance differential cross section (60) on the positron outgoing angle (for Channel A) or electron outgoing angle (for Channel B) for various frequencies and intensities, as well as the numbers of absorbed and emitted photons at the first and second vertices (see Figure 2). The study focused on the regions of optical and X-ray frequencies of the external strong electromagnetic wave (31) at different sufficiently high energies of initial gamma quanta. It is important to note that the energy of the second high-energy gamma quantum for each frequency and intensity of the wave was chosen according to condition (33) in order for the stimulated Breit–Wheeler process to occur with the highest probability, and the energy of the first gamma quantum was chosen to be much lower than the energy of the second gamma quantum (37). In this case, with the increasing frequency of the external field, the characteristic energy of the Breit–Wheeler process decreased (see relation (31)). Therefore, the energies of the initial gamma quanta were chosen to be lower for the X-ray frequency range than for the optical

frequency range. As a result, the function (61) increased, leading to an increase in the maximum resonance cross section. This case is shown in Figure 6a–c. However, when the energy of initial gamma quanta remained constant and the intensity of the external field increased, then the maximum resonance cross section decreased (see Figure 6c,c'). Table 1 displays the values of the positron (for Channel A) and electron (for Channel B) energies, as well as the corresponding maximum values of the resonance differential cross section according to their spectral-angular distribution (see Figure 6a–c') for different frequencies and intensities of the wave and the different energies of initial the gamma quanta.

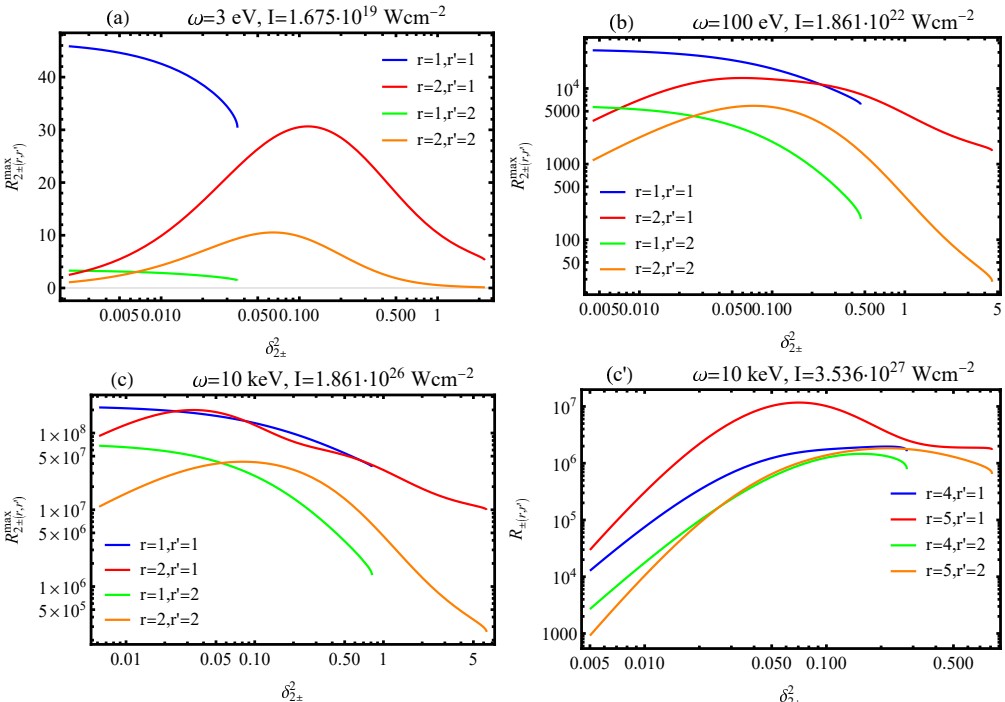

**Figure 6.** The dependence of the maximum resonance differential cross section (60) (in units of $r_e^2$) on the ultrarelativistic parameter $\delta_{2+}^2$ for (Channel A) or $\delta_{2-}^2$ (Channel B) for various frequencies and intensities, as well as the numbers of absorbed ($r$) and emitted ($r'$) photons. The value of the initial ultrarelativistic parameter was $\delta_{\eta i}^2 = 10^{-4}$. The energies of the initial gamma quanta were the following: (**a**) $\omega_1 = 10\,\text{GeV}, \omega_2 = 180\,\text{GeV}$; (**b**) $\omega_1 = 0.5\,\text{GeV}, \omega_2 = 7\,\text{Gev}$; and (**c,c'**) $\omega_1 = 10\,\text{MeV}, \omega_2 = 80\,\text{MeV}$.

From Table 1, it can be observed that when the energy of one of the initial gamma quanta slightly exceeded the characteristic Breit–Wheeler energy, the production of electron–positron pairs occurred with a very large cross section. For the optical frequency range, the resonance differential cross section could exceed the value of $r_e^2$ in magnitude by a factor of 47, while, for the X-ray frequency range, it could exceed the value of $r_e^2$ by eight orders of magnitude. In this case, the positrons (electrons) were emitted in a narrow cone and with very high energy.

We would like to emphasize that the article considered the circular polarization of an external electromagnetic wave. For this polarization, there was an azimuthal symmetry in the problem. We also emphasize that unpolarized gamma quanta were considered in the article. The study of this resonant process for the linear polarization of a wave or randomly polarized waves will significantly affect the angular distribution of an electron-positron pair [33–35,39,40,42]. At the same time, the effect of polarization of the initial gamma quanta on the Breit–Wheeler resonant process in strong fields is of undoubted interest and is an independent task that will be studied in future publications.

**Table 1.** The values of the ultrarelativistic parameter $\delta_{2\pm}^{2(*)}$ and the resonant energies of the positron (electron) $x_{\pm(r)}^{(*)}$ that corresponded to the maximum values $R_{2\pm(rr')}^{max(*)}$ (in units of $r_e^2$) of the spectral–angular distribution for the resonant differential cross sections (60) (see Figure 6).

| | $(r, r')$ | $\delta_{2\pm}^{2(*)}$ | $x_{\pm(r)}^{(*)}$ | $R_{2\pm(rr')}^{max(*)}$ |
|---|---|---|---|---|
| $I = 1.675 \times 10^{19}\,\mathrm{Wcm}^{-2}$, $\omega = 3\,\mathrm{eV}$, $\omega_1 = 10\,\mathrm{GeV}$, $\omega_2 = 180\,\mathrm{GeV}$ | (1,1) | 0 | 0.56 | 47 |
| | (1,2) | 0 | | 3 |
| | (2,1) | 0.12 | 0.76 | 31 |
| | (2,2) | 0.06 | 0.78 | 11 |
| $I = 1.861 \times 10^{22}\,\mathrm{Wcm}^{-2}$, $\omega = 100\,\mathrm{eV}$, $\omega_1 = 0.5\,\mathrm{GeV}$, $\omega_2 = 7\,\mathrm{GeV}$ | (1,1) | 0 | 0.70 | $3.3 \times 10^4$ |
| | (1,2) | 0 | | $6.0 \times 10^3$ |
| | (2,1) | 0.06 | 0.82 | $1.4 \times 10^4$ |
| | (2,2) | 0.07 | 0.81 | $5.9 \times 10^3$ |
| $I = 1.861 \times 10^{26}\,\mathrm{Wcm}^{-2}$, $\omega = 10\,\mathrm{keV}$, $\omega_1 = 10\,\mathrm{MeV}$, $\omega_2 = 80\,\mathrm{MeV}$ | (1,1) | 0 | 0.71 | $2.3 \times 10^8$ |
| | (1,2) | 0 | | $7.4 \times 10^7$ |
| | (2,1) | 0.03 | 0.8 | $2.0 \times 10^8$ |
| | (2,2) | 0.08 | 0.79 | $4.2 \times 10^7$ |
| $I = 1.675 \times 10^{27}\,\mathrm{Wcm}^{-2}$, $\omega = 10\,\mathrm{keV}$, $\omega_1 = 10\,\mathrm{MeV}$, $\omega_2 = 80\,\mathrm{MeV}$ | (4,1) | 0.22 | 0.47 | $2.0 \times 10^6$ |
| | (4,2) | 0.16 | 0.5 | $1.5 \times 10^6$ |
| | (5,1) | 0.07 | 0.64 | $1.2 \times 10^7$ |
| | (5,2) | 0.22 | 0.56 | $1.8 \times 10^6$ |

## 5. Conclusions

We considered the resonant Breit–Wheeler process modified by an external strong electromagnetic field for high-energy initial gamma quanta when the energy of one of them significantly exceeded the energy of the other. The following results were obtained:

1. The resonant kinematics of the process were studied in detail. It was demonstrated that the problem involves two characteristic energies: the Breit–Wheeler energy $\omega_{BW}$ (31) and the Compton effect energy $\omega_C$ (36). These energies differed from each other by a factor of four. The ratios of the initial gamma quanta energies to these characteristic energies significantly affected the number of absorbed or emitted wave photons and, ultimately, the probability of the process.

2. The resonant energies of the positron and electron strongly depended on their outgoing angles, as well as the characteristic quantum parameters $\varepsilon_{2BW(r)}$ (30) and $\varepsilon_{1C(r')}$ (36). Furthermore, the outgoing angles of the electron and positron were deemed to be interdependent (39).

3. The maximum resonant differential cross section was achieved when the energy of one of the initial gamma quantum slightly exceeded the characteristic Breit–Wheeler energy. In this case, for the optical frequency range and $\omega_2 = 180\,\mathrm{GeV}$, the maximum resonant cross section was $R_{2\pm(rr')}^{max} = 47\,r_e^2$, whereas, for the X-ray frequency range, it was $R_{2\pm(rr')}^{max} \sim (10^6 \div 10^8)\,r_e^2$.

The obtained results can be utilized to achieve ultrarelativistic positron (electron) beams with a very high probability in external field-modified Breit–Wheeler processes. Additionally, these results can be employed to explain the fluxes of ultrarelativistic positrons (electrons) near neutron stars and magnetars [59,60], as well as in the modeling of physical processes involving laser-induced thermonuclear fusion [61].

**Author Contributions:** Conceptualization, S.P.R.; methodology, S.P.R. and V.V.D.; software, V.D.S.; validation, S.P.R., V.V.D. and V.D.S.; formal analysis, S.P.R. and V.D.S.; investigation, S.P.R., V.V.D. and V.D.S.; resources, V.V.D.; data curation, S.P.R. and V.V.D.; writing—original-draft preparation, V.D.S.; writing—review and editing, S.P.R. and V.D.S.; visualization, V.D.S.; supervision, S.P.R. and V.V.D.; project administration, V.V.D.; funding acquisition, V.V.D. All authors have read and agreed to the published version of the manuscript.

**Funding:** The research was funded by the Ministry of Science and Higher Education of the Russian Federation under the strategic academic leadership program "Priority 2030" (Agreement 075-15-2023-380 dated 20 February 2023).

**Institutional Review Board Statement:** Not applicable.

**Informed Consent Statement:** Not applicable.

**Data Availability Statement:** The data presented in this study are openly available.

**Conflicts of Interest:** The authors declare no conflict of interest.

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
