# Peer review of "Generation of Narrow Beams of Ultrarelativistic Positrons (Electrons) in the Breit–Wheeler Resonant Process Modified by the Field of a Strong Electromagnetic Wave"

_photonics, doi:10.3390/photonics10080949_

Round 1
Reviewer 1 Report
The manuscript describes e+e- pair production when two gamma photons (of largely different energies) collide under a very narrow angle in the presence of a strong background laser wave of circular polarization. It is shown that the process proceeds in a resonant way, where a
Breit-Wheeler effect is followed by Compton scattering, and that it leads to the emission of a collimated beam of ultrarelativistic positrons (or electrons). The manuscript is well written and clearly structured.
In principle, the paper deserves to be published - provided the authors properly address the following remarks in a revised version of the paper:
1) The important assumption that the collision angle between the gamma photons is very narrow, is a bit hidden in Eq.(24). This assumption should be stated more clearly. The best would be to show a schematic picture of the beam configuration.
2) Breit-Wheeler pair production (nonresonant) by two high-frequency photons in the presence of a low-frequency field has also been considered in T. Nousch et al., Phys. Lett. B 755, 162 (2016) and M. Jansen et al., Phys. Rev. A 88, 052125 (2013). These closely related studies should be mentioned.
3) Generation of highly collimated ultrarelativistic positron beams by laser-driven pair production has also been studied in H. Chen et al., Phys. Rev. Lett. 105, 015003 (2010); G. Sarri et al., Phys. Rev. Lett. 110, 255002 (2013); F. Wan et al., Phys. Lett. B 800, 135120 (2019). These related papers deserve to be mentioned, as well.
4) Currently there are concrete plans in various strong-field laboratories to measure the nonlinear Breit-Wheeler effect. The authors might want to mention these activities and add corresponding references to strengthen the motivation and to show the topicality of their study: e.g. I. Turcu et al., Rom. Rep. Phys. 68, S145 (2016); H. Abramowicz et al., Eur. Phys. J. Spec. Top. 230, 2445 (2021); F. Salgado et al., New J. Phys. 23, 105002 (2021).
5) The physical meaning of omega_C in Eq.(36) should be explained. While the meaning of omega_BW is clear, since pair production has a threshold, for omega_C it is not so clear.
6) Below Eq.(37) the authors say that the channels A and B are distinguishable and do not interfere. This statement sounds too strong because in general the channels do interfere. The interference is, however, very small for the setup under consideration.
7) Below Eq.(64) the authors say that the resonant cross section exceeds the laser field-free cross section significantly. This statement should be elaborated in more detail (because it is not easily seen from the formulas) and should also be quantified: how much larger is the resonant cross section?
In addition, some minor points:
8) The angle \theta_\pm in Eq.(26) needs to be defined. This symbol has not been introduced in Eqs.(24), (25).
9) The axis labels in Fig.3 are very small.
10) A reference should be given where a reader can find Eqs.(52)-(55) for nonlinear Compton scattering.
11) It should be stated in Table 1 that the quantity R^max in the last column is given in units of r_e^2. And in the paragraph below Table 1 the wording 'exceed the value' needs to be precised (which value?).
As far as I can judge, the English language of the manuscript is fine and requires only some minor editing.
Reviewer 2 Report
The article entitled, "Generation of narrow beams of ultrarelativistic positrons (electrons) in the Breit-Wheeler resonant process, modified by the field of a strong electromagnetic wave" follows the line of work of these authors in several previous manuscripts. In this article the authors investigate the resonant Breit-Wheeler process in a strong electromagnetic field. In this article the differential resonant cross sections are investigated with special attention given to the angular dependences.
The article is generally well organized and clearly written. The results appear to be reasonable based on other similar work and provide additional information compared to previous publications. The introduction, while complete is extremely similar to that of previous manuscripts by the same authors. The current manuscript would be improved by a paragraph or two emphasizing the differences and advances made specifically in this manuscript. Since the angular distributions are impacted by the initial photon polarization I would like to see some additional discussion of the photon polarization impact. The article addresses circularly polarized waves. It would be useful to at least comment on the modifications for linearly polarized or randomly polarized waves as this would more closely match some potential experimental conditions.
lines 106-109: These angles are not well defined / explained. It took me some time thinking through this and looking at past papers by these authors to understand what is meant here. I suggest to improve this by spending a few lines defining / discussing the meaning of $\theta_{j\pm}$, $\theta_{i}$, and $\theta$, $\theta_{j}$. Further, I am not sure I understand what is meant by ... $\sim1$, is this in units of $\pi$, should it instead be $\sim\pi$? I'd like to hear the authors comments and justification on this point.
Around lines 86 and 110: the terms "quasi" momentum and energy are used. What do the authors mean in exactly and what approximation is referred to here?
In Eq 12-15 the "r" variables are undefined in this manuscript. I assume they are similarrly defined as in [9], [30] but it should be made clear in this manuscript.
lines 123-124: Please specify the frequencies used exactly instead of only giving there approximate (optical or X-ray).
Can the authors expound upon this statement: "Therefore, Channels A and B are distinguishable and do not interfere 177 with each other." It is not clear to me why the previous statement leads necessarily to the distinguishability of the amplitudes and therefore no interference. Since this argument is refereed to later it is worth making especially clear.
Figure 4: shows limited domain of each of these curves, can the authors comment on this and point to or make more clear which conditions limits the domain of each curve.
Figure 5: I dont understand what is gained with the unintutive log scale on the x-axis (angle). Make this linear or at least make the tick marks correspond to fractions of pi instead of being decimals.
Similarly, please add one or two sentences discussing why the curves in Fig 5 with r=1 or r'=1 end where they do (make it clear).
In the conclusions you state that: "Furthermore, the outgoing angles of the electron and positron are interdependent" - could you comment more on this, it it simply due to momentum resolution or are there other conservation effects also dictating this (i.e. spin)?
Please add a suitable reference for this statement in the conclusion: "ultrarelativistic positrons 320 (electrons) near neutron stars and magnetars" and "as well as in the modeling of physical pro- 321 cesses in laser-induced thermonuclear fusion"
General:
Add a space between numerals and units throughout.
Please verify that all variables and notations are clearly defined in this manuscript
In several places you mention optical vs. x-ray frequency ranges. Please simply add the numerical value of the range you are utilizing for these.
